# Dietary Supplementation with Chitosan Oligosaccharides Alleviates Oxidative Stress in Rats Challenged with Hydrogen Peroxide

**DOI:** 10.3390/ani10010055

**Published:** 2019-12-26

**Authors:** Ruixia Lan, Qingqing Chang, Lilong An, Zhihui Zhao

**Affiliations:** Department of Animal Science, College of Agriculture, Guangdong Ocean University, Zhanjiang 524-088, China; Lanrx@gdou.edu.cn (R.L.); changqingqing@outlook.com (Q.C.); anlilong_gdou@163.com (L.A.)

**Keywords:** chitosan oligosaccharides, hydrogen peroxide, oxidative stress, rat

## Abstract

**Simple Summary:**

Oxidative stress adversely affects animal health and performance. Feed additives with antioxidant abilities supplementation can alleviate oxidative stress. The aim of this study was to evaluate the hypothesis that dietary supplementation with COS alleviates the damage caused by oxidative stress in Sprague Dawley rats challenged with hydrogen peroxide (H_2_O_2_). The results shown that COS exhibited better radical scavenging capacity of 1, 1-diphenyl-2-picrylhydrazyl (DPPH), superoxide anion (O_2_^−^), H_2_O_2_, and ferric ion reducing antioxidant power (FRAP) than butylated hydroxy anisole (BHA), increasing activity of SOD, CAT, GSH-Px, GSH, and T-AOC, as well as decreasing MDA level in serum, liver, spleen, and kidney. Our results indicated that COS can protect Sprague Dawley rats from H_2_O_2_ challenge by reducing lipid peroxidation and restoring antioxidant capacity.

**Abstract:**

Oxidative stress is induced by excessive oxidative radicals, which directly react with biomolecules, and damage lipids, proteins and DNA, leading to cell or organ injury. Supplementation of antioxidants to animals can be an effective way to modulate the antioxidant system. Chitosan oligosaccharides (COS) are the degraded products of chitosan or chitin, which has strong antioxidant, anti-inflammatory, and immune-enhancing competency. Therefore, the current study was conducted to evaluate the hypothesis that dietary supplementation with COS alleviates the damage caused by oxidative stress in Sprague Dawley rats challenged with hydrogen peroxide (H_2_O_2_). The rats were randomly divided into three groups: CON, control group, in which rats were fed a basal diet with normal drinking water; AS, H_2_O_2_ group, in which rats were fed the basal diet and 0.1% H_2_O_2_ in the drinking water; ASC, AS + COS group, in which rats were fed the basal diet with 200 mg/kg COS, and with 0.1% H_2_O_2_ in the drinking water. In vitro, COS exhibited better radical scavenging capacity of 1, 1-diphenyl-2-picrylhydrazyl (DPPH), superoxide anion (O_2_^−^), H_2_O_2_, and ferric ion reducing antioxidant power (FRAP) than butylated hydroxy anisole (BHA). In vivo, dietary supplementation with COS alleviated the H_2_O_2_-induced oxidative damage, evidenced by comparatively increasing activity of SOD, CAT, GSH-Px, GSH, and T-AOC, and comparatively decreasing level of MDA in serum, liver, spleen, and kidney. COS also comparatively alleviated the H_2_O_2_-induced inflammation. In conclusion, COS supplementation reduced lipid peroxidation and restored antioxidant capacity in Sprague Dawley rats, which were challenged with H_2_O_2_.

## 1. Introduction

Oxidative stress is induced by excessive oxidative radicals, which directly react with biomolecules, damaging lipids, proteins, and DNA, and leading to cell or organ injury [1,2,3,4]. Moreover, oxidative stress disturbs the normal intestinal barrier function, decreases the expression of tight junction proteins, and reduces nutrient absorption and metabolism, resulting in growth depression [5,6,7,8,9]. Oxidative stress is also associated with health disorders, such as inflammatory disease, diabetes, and cancer [10,11]. During the past decades, evaluation of free radical generation and antioxidant defense have been widely investigated in animals [12]. Numerous studies reported that dietary antioxidant supplementation, such as Vitamin C, Vitamin E, curcumin, and *Forsythia suspense* extract, given to animals is a useful way to alleviate oxidative stress [13,14,15,16]. Thus, dietary antioxidant supplementation may be a useful way to modulate the antioxidant system. Chitosan oligosaccharides (COS) are the degraded products of chitosan or chitin [17], which can be used as a natural antioxidants [18] due to its strong antioxidant competency [19,20]. Furthermore, COS exhibit other biological activities, such as anti-bacterial [21,22], anti-inflammatory [23,24,25], and immune-enhancing activities [26,27]. The aim of this study was to evaluate the hypothesis that dietary supplementation with COS alleviates the oxidative stress in Sprague Dawley rats challenged with hydrogen peroxide (H_2_O_2_).

## 2. Materials and Methods

### 2.1. Animal Ethics

The experimental protocol used in this study was approved by the Animal Care and Use Committee of Guangdong Ocean University, China (SYXK-2018-0147).

### 2.2. Chemicals and Reagents

Chitosan oligosaccharides were purchased from Jiangsu Xinrui Biotechnology Co., Ltd. (HPLC purity 95%, deacetylation degree over 95% and average molecular weight below 32 kDa; Xuzhou, Jiangsu, China). We purchased 1, 1-diphenyl-2-picrylhydrazyl (DPPH), phenazine methosulfate (PMS), nicotinamide adenine dinucleotide (NADH), nitroblue tetrazolium (NBT), 2, 4, 6-tri (2-pyridyl)-s-triazine (TPTZ), 2, 2′-azobis (2-amidinopropane) dihydrochloride (AAPH), and butylated hydroxy anisole (BHA) from Sigma-Aldrich (Shanghai, China).

The kits used for analyzing the activity/level of superoxide dismutase (SOD), catalase (CAT), glutathione peroxidase (GSH-Px), glutathione (GSH), total antioxidant capacity (T-AOC), the content of malondialdehyde (MDA), interleukin-1β (IL-1β), interleukin-6 (IL-6), interleukin-10 (IL-10), tumor necrosis factor-α (TNF-α), and BCA protein were provided by Nanjing Jiancheng Bioengineering Institute (Nanjing, Jiangsu, China). Other chemicals were from Shanghai chemical agents’ company, China.

### 2.3. DPPH, O_2_^−^, H_2_O_2_, and FRAP Radical Scavenging Capacity

The DPPH, superoxide anion (O_2_^−^), H_2_O_2_, and ferric ion reducing antioxidant power (FRAP) radical scavenging capacity of COS and BHA were measured by the method described by Hou et al. [28]. Briefly, DPPH was dissolved in ethanol to a 0.1 mM solution. The sample and DPPH at 0.01, 0.02, 0.03, 0.04, 0.05, and 0.06 g/L were mixed vigorously. After incubation at room temperature in the darkness for 30 min, the absorbance was determined at 517 nm. The scavenging effect was calculated by the equation: DPPH scavenging ability (%) = (A_control_ − A_sample_)/A_control_ × 100%. For analyzing the O_2_^−^ scavenging activity, PMS, NADH and NBT were dissolved in phosphate buffer (PBS) to a 60,468 and 150 μM solution, respectively. The reaction mixture included 1 mL of sample solution at 0.01, 0.02, 0.03, 0.04, 0.05, and 0.06 g/L, and 1 mL of PMS, NADH, and NBT solutions, and was then incubated for 5 min at room temperature and the absorbance was determined at 560 nm. The scavenging effect was calculated by the equation: O_2_^−^ scavenging ability (%) = (A_control_ − A_sample_)/A_control_ × 100%. For analyzing H_2_O_2_ scavenging activity, a 43 mM H_2_O_2_ solution was prepared in PBS. Then 3.4 mL of sample solutions at 0.01, 0.02, 0.03, 0.04, 0.05, and 0.06 g/L was mixed with 0.6 mL of a 43 mM H_2_O_2_ solution, and the absorbance was determined at 230 nm. The scavenging effect was calculated by the equation: H_2_O_2_ scavenging ability (%) = (A_control_ − A_sample_)/A_control_ × 100%. For FRAP analyzing, the FRAP reagent consisted of acetate buffer (0.3 M, pH = 3.6), TPTZ (10 mM) in hydrochloric acid (40 mM), and ferric chloride (20 mM), at a ratio of 10:1:1. The FRAP reagent was prepared just before the reaction, then 1.5 mL of FRAP reagent was mixed with 150 μL distilled water and 50 μL sample solutions at 0.01, 0.02, 0.03, 0.04, 0.05, and 0.06 g/L. It was then incubated for 30 min at room temperature in the darkness and the absorbance was determined at 593 nm. The reducing power is presented as micromoles Fe/gram fresh weight.

### 2.4. Animals, Experiment Design, and Diets

The male Sprague Dawley rats (8–10 weeks, 178.39 ± 5.12 g) used in this study were obtained from the Beijing Administration Office of Laboratory Animals (Beijing, China). The rats were individually housed in polycarbonate cages with soft wood granulate floors, and kept at 24 °C, with a 12 h light–dark cycle. After a week of acclimatization, 30 rats were randomly divided into one of 3 groups with 10 rats in each group for this 10 day experiment: CON, control group, rats were fed basal diet and normal drinking water; AS, H_2_O_2_ group, rats were fed basal diet with 0.1% H_2_O_2_ in drinking water [29]; ASC, AS + COS group, rats were fed basal diet with 200 mg/kg COS, and with 0.1% H_2_O_2_ in drinking water. The supplementation level of COS was based on our previous preliminary research. All rats had free access to water and diet. The composition of the basal diet is listed in Table 1, and was made according to the nutritional requirement recommendations by the American Institute of Nutrition-93 diet [30].

### 2.5. Plasma Collection and Tissue Preparation

On the last day of the experiment, after 12 h fasting, all rats were euthanized with diethyl ether. Blood was collected from the posterior vena orbitalis, then centrifuged at 3000× *g* × 10 min to collect the serum, and stored at −20 °C until analysis. The liver, kidney, and spleen were collected and a 10% homogenate was prepared in PBS and centrifuged at 3000× *g* × 10 min at 4 °C, the supernatant was used for further biochemical assays.

### 2.6. Antioxidant and Inflammatory Cytokines Assays

The content or the activity of MDA, SOD, CAT, GSH-Px, GSH, T-AOC, IL-1β, IL-6, IL-10, and TNF-α in serum, liver, kidney, and spleen were measured according to the manufacturer’s instruction.

### 2.7. Statistical Analysis

Data were analyzed by ANOVA using the GLM procedures of SAS (V9.1, SAS Inst., Inc., Cary, NC, USA). Duncan’s multiple range tests were done to check the differences among treatments, and *p* < 0.05 was considered significant.

## 3. Results

### 3.1. DPPH, O_2_^−^, H_2_O_2_, and FRAP Scavenging Capacity

The DPPH, O_2_^−^, H_2_O_2_, and FRAP scavenging capacity of COS and BHA is shown in Figure 1. The concentration ranges from 0.01 to 0.06 g/L, the scavenging activity of COS ranges from 50.19% to 61.77%, and the scavenging activity of BHA ranges from 24.52% to 36.10%. For O_2_^−^ scavenging capacity, the scavenging activity of COS ranges from 32.94% to 40.31%, and the scavenging activity of BHA ranges from 23.45% to 30.02%. For the H_2_O_2_ scavenging capacity, the scavenging activity of COS ranges from 61.65% to 86.21%, and the scavenging activity of BHA ranges from 38.07% to 49.10%. For ferric reducing antioxidant power (FRAP) scavenging capacity, the scavenging activity of COS ranges from 44.24% to 42.62%, and the scavenging activity of BHA ranges from 23.73% to 24.93%.

### 3.2. Effects of COS on Antioxidant Status in Serum

Administration of H_2_O_2_ in drinking water increased (*p* < 0.05) the content of MDA in serum (Figure 2A), and decreased (*p* < 0.05) the activity of SOD (Figure 2B) and CAT (Figure 2C) in the AS group compared with the CON group. No significant differences were observed on the activity of MDA, SOD, CAT, GSH-Px, GSH, or T-AOC.

### 3.3. Effects of COS on Antioxidant Status in the Liver

The activity of SOD and GSH in the AS group was significantly (*p* < 0.05) lower than that in the CON group (Figure 3B,E) in the liver. The activity of CAT (Figure 3C), GSH-Px (Figure 3D), and T-AOC (Figure 3F), and the content of MDA (Figure 3A) were not significantly affected by exposure to H_2_O_2_ or supplementation with COS.

### 3.4. Effects of COS on Antioxidant Status in the Spleen

The activity of CAT and GSH-Px in the AS group was significantly (*p* < 0.05) lower than that in the CON group (Figure 4C,D) in the spleen. However, the activity of SOD (Figure 4B), GSH (Figure 4E), and T-AOC (Figure 4F), and the content of MDA (Figure 4A), were not significantly affected by exposure to H_2_O_2_ or supplementation with COS.

### 3.5. Effects of COS on Antioxidant Status in the Kidney

The activity of SOD, GSH-Px, GSH, and T-AOC in the AS group was significantly (*p* < 0.05) lower than that in the CON group (Figure 5B–F, respectively), the activity of GSH and T-AOC in the AS group was also lower (*p* < 0.05) than that in the ASC group. The activity of CAT (Figure 5C) and the content of MDA (Figure 4A) in the kidney were not significantly affected by exposure to H_2_O_2_ or supplementation with COS.

### 3.6. Effects of COS on Inflammatory Cytokines in the Serum, Liver, Spleen, and Kidney

The content of IL-1β, IL-6, TNF-α, and IL-10 in the serum, liver, spleen, and kidney are presented in Figure 6, Figure 7, Figure 8 and Figure 9, respectively. The content of IL-1β (Figure 6A, Figure 7A, Figure 8A and Figure 9A), IL-6 (Figure 6B, Figure 7B, Figure 8B and Figure 9B), IL-10 (Figure 6C, Figure 7C and Figure 9C), and TNF-α (Figure 6D, Figure 7D, Figure 8D and Figure 9D) in the serum, liver, spleen, and kidney did not differ among treatments, except the content of IL-10 (Figure 8C) in the spleen. The amount of IL-10 was significantly lower (*p* < 0.05) in the AS group compared with the CON group.

## 4. Discussion

Increasing attention has been paid to the field of antioxidants in recent years. Food scientists have paid attention to antioxidants because of their ability to prevent fat from oxidative rancidity. Doctors are interested because of their ability to protect from oxidative injury. COS possess stronger antioxidant competency as evidenced by their reaction with unstable free radicals to form stable radicals [31]. In the present study, the effects of COS on antioxidant property were analyzed both in vitro and in vivo.

DPPH is a stable nitrogen radical, and is widely used to evaluate radical quenching capacities. The DPPH scavenging ability of COS was higher than BHA. H_2_O_2_, and O_2_^−^ act as signaling intermediates, and the over-production of H_2_O_2_ and O_2_^−^ is an indicator of oxidative stress [32]. DPPH reflects limited oxidation situations because it only exists in vitro; therefore, we further evaluated the H_2_O_2_ and O_2_^−^ scavenging capacity of COS. H_2_O_2_ can be synthesized and destroyed in response to external stimuli [33], and O_2_^−^ acts as a potential precursor to generate reactive radical species. Thus, evaluating the scavenging capacity of H_2_O_2_ and O_2_^−^ [34] is important for clarifying the antioxidant capacity. The results showed that the scavenging capacity of COS against H_2_O_2_ and O_2_^−^ was higher than BHA. There is an association between antioxidant capacity of antioxidants and their reducing power. We investigated the reducing power of COS using the FRAP assay. The results showed the scavenging capacity of COS against FRAP was higher than BHA. The antioxidant capacity of COS is related to its characteristic structure, including its deacetylation ratio, molecular weight, and the source of the material. The molecular weight and deacetylation ratio of COS also exerts some synergistic effects on the biological capacities, where, generally, the low molecular weight gives the stronger scavenging activity of DPPH, H_2_O_2_ and O_2_^−^ [35].

Many reports indicated that oxidative stress was related to animal health [36]. H_2_O_2_ is generally used as an oxidative stress stimulus, and studies have indicated that oral administration or intraperitoneal administration of H_2_O_2_ induces oxidative stress [37,38]. The detrimental effects of H_2_O_2_ depends on its conversation to hydroxy ions and other subsequent redox products. In this study, H_2_O_2_ challenge increased the content of MDA in the serum, liver, and kidney. Increasing content of MDA is an indicator of lipid peroxidation. COS supplementation resulted in a comparative decrease in MDA content in the serum, liver, and kidney, indicating that COS had protective effects due to its antioxidant capacity [39].

Both SOD and CAT can scavenge superoxide ions and hydroxyl ions. The present study showed decreasing activity of SOD in the serum, liver and kidney, and decreasing activity of CAT in the serum and spleen of H_2_O_2_-exposed rats. H_2_O_2_ induced a dramatic decrease in the activity of SOD and CAT, which may be related to the formation of reactive oxygen species (ROS) [29]. Dietary COS increased the activity of SOD and CAT, which can save the depletion of the two enzymes. GSH-Px is a glutathione-related enzyme, and there was a significant decrease in the activity of GSH-Px in the spleen and kidney of H_2_O_2_-exposed rats in this study. H_2_O_2_ can efficiently be scavenged by GSH-Px; therefore, the decreasing activity of GSH-Px reflects perturbations in the normal oxidative balance by H_2_O_2_ exposure. GSH is a typical non-enzyme antioxidant, which defends against reactive free radicals and other oxidant species in cellular defense systems. In this study, there was a significant decrease in the activity of GSH in the liver and kidney of H_2_O_2_-exposed rats, indicating that the depletion of GSH led to enhanced formation of ROS [40]. Meanwhile, COS supplementation increased the activity of GSH in the kidney, which suggests that COS maintained the redox balance of ROS through both the enzymatic and non-enzymatic antioxidant defense system. The value of T-AOC reflects the total antioxidant capacity [41]. In this study, the T-AOC value of the kidney decreased by H_2_O_2_ exposure, but increased with COS supplementation, which suggests that COS supplementation can alleviate oxidative stress induced by H_2_O_2_ through the non-enzymatic antioxidant system in the kidney. Both in vitro and in vivo experiments showed that COS has strong radical scavenging activity and antioxidant capacity. The radical scavenging activity of COS is associated with their proton donation ability [42]. Dietary supplementation with COS prevented H_2_O_2_-induced lipid peroxidation and reserved depletion of SOD, CAT, GSH-Px, and GSH activity, which was consistent with former reports indicating antioxidant and protective properties of COS [43,44,45].

Oxidative stress and inflammation are highly related [46]. Oxidative stress and excessive production of ROS is associated with inflammation, leading to the synthesis and release of pro-inflammatory cytokines. No significant differences were detected in the concentration of TNF-α, IL-1β, IL-6, and IL-10 in the serum, liver, spleen and kidney, except a significant decreasing concentration of IL-10 in the AS group compared to the CON group in the spleen. With COS supplementation, the level of TNF-α, IL-1β, and IL-6 in the ASC group was comparatively lower than that in the AS group in the serum, liver, spleen, and kidney, while the level of IL-10 in the ASC group was comparatively higher than that in the AS group, which may be due to the anti-inflammatory activity of COS [24].

## 5. Conclusions

In conclusion, COS had higher antioxidant activities than BHA when checked by DPPH, O_2_^−^, H_2_O_2_, and FRAP scavenging capacity in vitro. COS significantly increased the content of GSH and T-AOC in the kidney, and comparatively decreased the content of MDA in the serum, liver and kidney, which suggested COS had protective effects against H_2_O_2_-induced oxidative damage and can be used as a potential antioxidant in feed.

## Figures and Tables

**Figure 1 animals-10-00055-f001:**
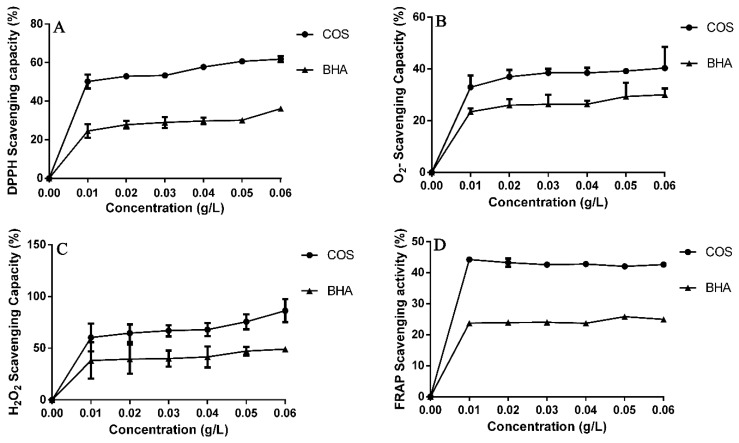
DPPH, O_2_^−^, H_2_O_2_, and FRAP radical scavenging capacity of COS and BHA at 0.01–0.06 g/L. Values expressed as mean ± standard error of three parallel experiments. (**A**) DPPH scavenging capacity; (**B**) O_2_^−^ scavenging capacity; (**C**) H_2_O_2_ scavenging capacity; (**D**) FRAP scavenging capacity.

**Figure 2 animals-10-00055-f002:**
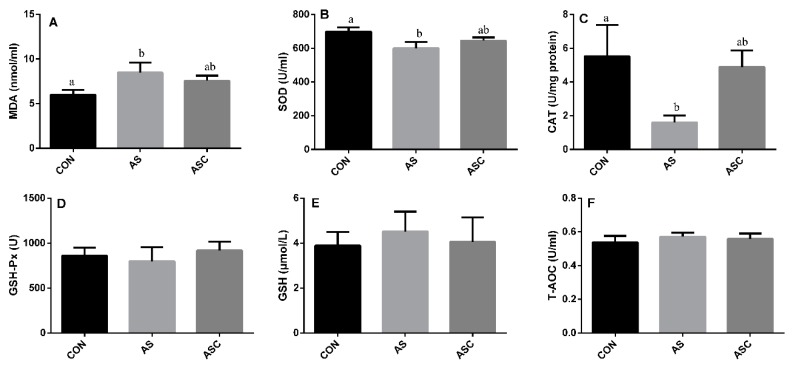
Effects of COS on antioxidant status in serum. Values expressed as mean ± standard error, n = 10. ^a,b^ Different superscript letters means significant differences (*p* < 0.05). CON, basal diet with normal drinking water; AS, basal diet with 0.1% H_2_O_2_ in drinking water; ASC, basal diet supplemented with 200 mg/kg COS, and with 0.1% H_2_O_2_ in drinking water. MDA, malondialdehyde; SOD, superoxide dismutase; CAT, catalase; GSH-Px, glutathione peroxidase; GSH, glutathione; T-AOC, total antioxidant capacity. (**A**) the content of MDA; (**B**) the activity of SOD; (**C**) the activity of CAT; (**D**) the activity of GSH-Px; (**E**) the content of GSH; (**F**) the content of T-AOC.

**Figure 3 animals-10-00055-f003:**
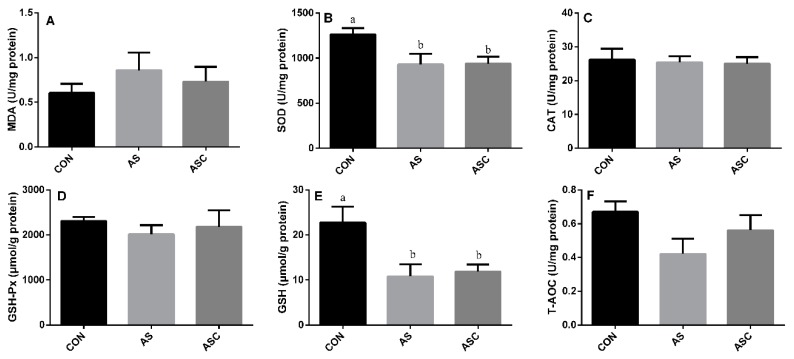
Effects of COS on antioxidant status in the liver. Values expressed as mean ± standard error, n = 10. ^a,b^ Different superscript letters mean significant differences (*p* < 0.05). CON, basal diet with normal drinking water; AS, basal diet with 0.1% H_2_O_2_ in drinking water; ASC, basal diet supplemented with 200 mg/kg COS, and with 0.1% H_2_O_2_ in drinking water. MDA, malondialdehyde; SOD, superoxide dismutase; CAT, catalase; GSH-Px, glutathione peroxidase; GSH, glutathione; T-AOC, total antioxidant capacity. (**A**) the content of MDA; (**B**) the activity of SOD; (**C**) the activity of CAT; (**D**) the activity of GSH-Px; (**E**) the content of GSH; (**F**) the content of T-AOC.

**Figure 4 animals-10-00055-f004:**
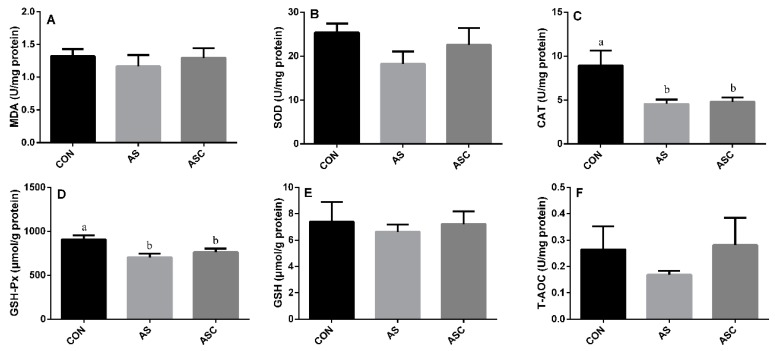
Effects of COS on antioxidant status in the spleen. Values expressed as mean ± standard error, n = 10. ^a,b^ Different superscript letters mean significant differences (*p* < 0.05). CON, basal diet with normal drinking water; AS, basal diet with 0.1% H_2_O_2_ in drinking water; ASC, basal diet supplemented with 200 mg/kg COS, and with 0.1% H_2_O_2_ in drinking water. MDA, malondialdehyde; SOD, superoxide dismutase; CAT, catalase; GSH-Px, glutathione peroxidase; GSH, glutathione; T-AOC, total antioxidant capacity. (**A**) the content of MDA; (**B**) the activity of SOD; (**C**) the activity of CAT; (**D**) the activity of GSH-Px; (**E**) the content of GSH; (**F**) the content of T-AOC.

**Figure 5 animals-10-00055-f005:**
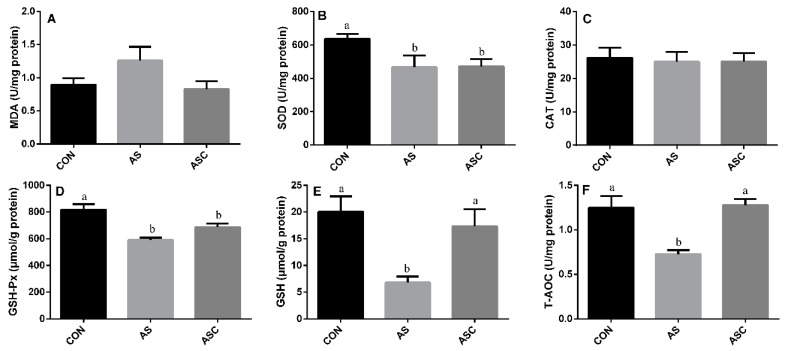
Effects of COS on antioxidant status in kidney. Values expressed as mean ± standard error, n = 10. ^a,b^ Different superscript letters mean significant differences (*p* < 0.05). CON, basal diet with normal drinking water; AS, basal diet with 0.1% H_2_O_2_ in drinking water; ASC, basal diet supplemented with 200 mg/kg COS, and with 0.1% H_2_O_2_ in drinking water. MDA, malondialdehyde; SOD, superoxide dismutase; CAT, catalase; GSH-Px, glutathione peroxidase; GSH, glutathione; T-AOC, total antioxidant capacity. (**A**) the content of MDA; (**B**) the activity of SOD; (**C**) the activity of CAT; (**D**) the activity of GSH-Px; (**E**) the content of GSH; (**F**) the content of T-AOC.

**Figure 6 animals-10-00055-f006:**
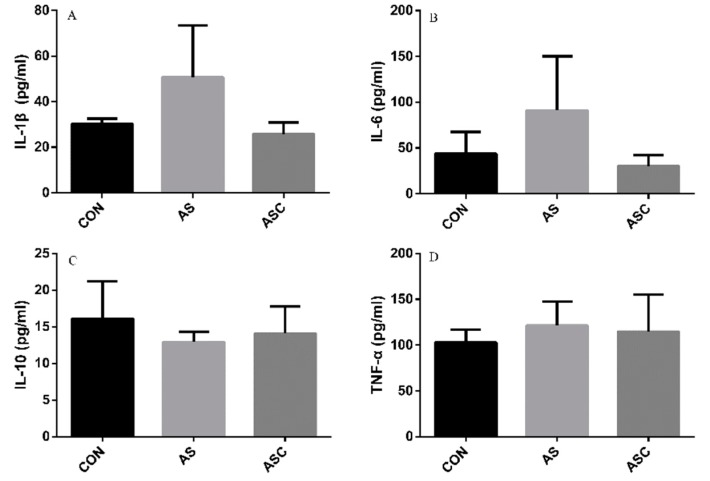
Effects of COS on inflammatory cytokines in serum. Values presented as mean ± standard error, n = 10. CON, basal diet with normal drinking water; AS, basal diet with 0.1% H_2_O_2_ in drinking water; ASC, basal diet supplemented with 200 mg/kg COS, and with 0.1% H_2_O_2_ in drinking water. IL-1β, interleukin-1β; IL-6, interleukin-6; IL-10, interleukin-10; TNF-α, tumor necrosis factor-α. (**A**) the concentration of IL-1β; (**B**) the concentration of IL-6; (**C**) the concentration of IL-10; (**D**) the concentration of TNF-α.

**Figure 7 animals-10-00055-f007:**
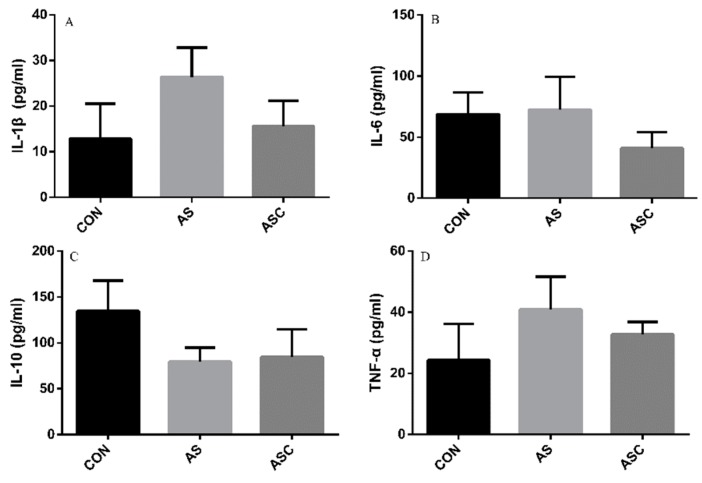
Effects of COS on inflammatory cytokines in the liver. Values presented as mean ± standard error, n = 10. CON, basal diet with normal drinking water; AS, basal diet with 0.1% H_2_O_2_ in drinking water; ASC, basal diet supplemented with 200 mg/kg COS, and with 0.1% H_2_O_2_ in drinking water. IL-1β, interleukin-1β; IL-6, interleukin-6; IL-10, interleukin-10; TNF-α, tumor necrosis factor-α. (**A**) the concentration of IL-1β; (**B**) the concentration of IL-6; (**C**) the concentration of IL-10; (**D**) the concentration of TNF-α.

**Figure 8 animals-10-00055-f008:**
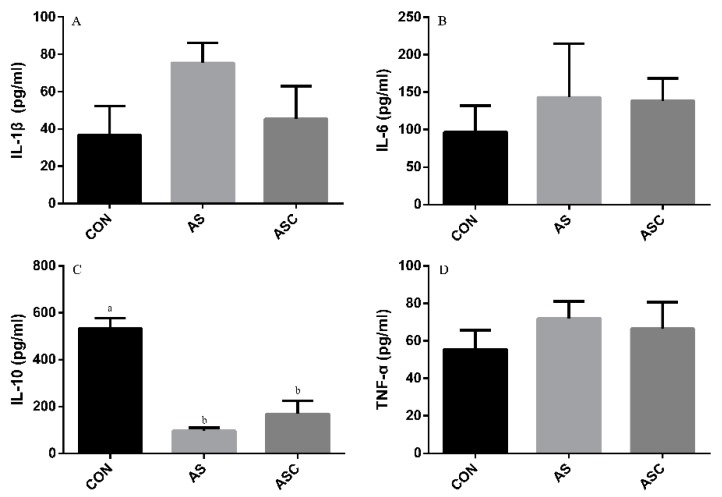
Effects of COS on inflammatory cytokines in the spleen. Values presented as mean ± standard error, n = 10. ^a,b^ Different superscript letters mean significant differences (*p* < 0.05). CON, basal diet with normal drinking water; AS, basal diet with 0.1% H_2_O_2_ in drinking water; ASC, basal diet supplemented with 200 mg/kg COS, and with 0.1% H_2_O_2_ in drinking water. IL-1β, interleukin-1β; IL-6, interleukin-6; IL-10, interleukin-10; TNF-α, tumor necrosis factor-α. (**A**) the concentration of IL-1β; (**B**) the concentration of IL-6; (**C**) the concentration of IL-10; (**D**) the concentration of TNF-α.

**Figure 9 animals-10-00055-f009:**
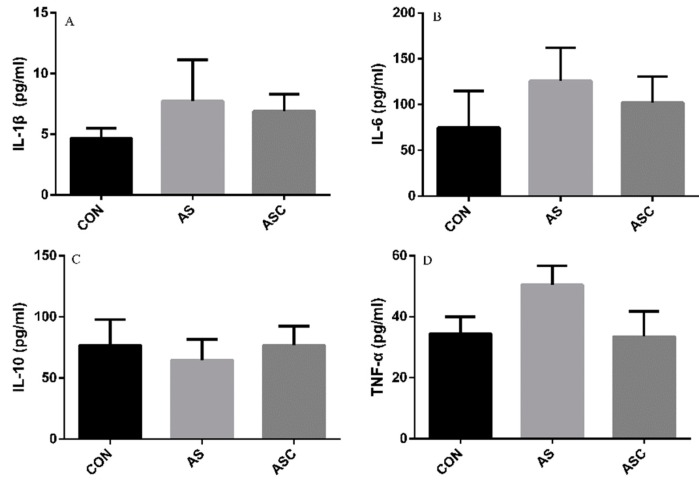
Effects of COS on inflammatory cytokines in the kidney. Values presented as mean ± standard error, n = 10. CON, basal diet with normal drinking water; AS, basal diet with 0.1% H_2_O_2_ in drinking water; ASC, basal diet supplemented with 200 mg/kg COS, and with 0.1% H_2_O_2_ in drinking water. IL-1β, interleukin-1β; IL-6, interleukin-6; IL-10, interleukin-10; TNF-α, tumor necrosis factor-α. (**A**) the concentration of IL-1β; (**B**) the concentration of IL-6; (**C**) the concentration of IL-10; (**D**) the concentration of TNF-α.

**Table 1 animals-10-00055-t001:** Dietary composition and nutrient content of the basal diet.

Ingredients, %	Basal Diet
Cornstarch	46.40
Casein	14.00
Dextrinized cornstarch	15.50
Sucrose	10.00
Soybean oil	4.00
Cellulose acetate	5.00
Mineral premix ^1^	3.50
Vitamin premix ^2^	1.00
L-Methionine	0.18
L-Cystine	0.18
Choline bitartrate	0.23
Tert-butylhydroquinone	0.01
Gross energy (MJ/kg)	16.22

^1^ Mineral premix (mg/kg of premix): CaCO_3_, 3.70 × 10^5^; KH_2_PO_4_, 1.96 × 10^5^; K_3_C_6_H_5_O_7_·H_2_O, 7.08 × 10^4^; NaCl, 7.4 × 10^4^; K_2_SO_4_, 4.66 × 10^4^; MgO, 2.4 × 10^4^; FeC_6_H_5_O_7_H_2_O, 6.06 × 10^3^; ZnCO_3_, 1.65 × 10^3^; MnCO_3_, 630; CuCO_3_, 324; NaSiO_3_·9H_2_O, 1.45 × 10^3^; CrK(SO_4_)12H_2_O, 275; LiCl, 17.4; H_3_BO_3_, 81.5; NaF, 63.5; NiCO_3_·2Ni(OH)_2_·4H_2_O, 30.6; NH_4_VO_3_, 6.6; sucrose was added to make a total of 1 kg.; ^2^ Vitamin premix (mg/kg of premix): Nicotinic, 3.0 × 10^3^; calcium pantothenate, 1.6 × 10^3^; pyridoxine hydrochloride, 700; thiamine hydrochloride, 600; riboflavin, 600; folic acid, 200; D-biotin, 20; cyanocobalamin, 2.5 × 10^3^; a-tocopherol, 1.5 × 10^4^; cholecalciferol, 250; phylloquinone, 75; sucrose was added to make a total of 1 kg.

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
