# Peer review of "Dietary Supplementation with Chitosan Oligosaccharides Alleviates Oxidative Stress in Rats Challenged with Hydrogen Peroxide"

_animals, 2019, doi:10.3390/ani10010055_

Round 1

Reviewer 1 Report

The manuscript describes about dietary supplementation of Chitosan oligosaccharide for having its ability to alleviate oxidative stress in rats that are exposed to hydrogen peroxide which is quite interesting.

The English of this paper needs corrections. Please have the whole paper checked again and corrected by an English-speaking person who is familiar with scientific writing.

Some of the sentences are started with abbreviations. Do not start the sentences with abbreviations.

Specific comments

Abstract

Line 13. Supplementation antioxidants – Supplementation of antioxidants

Line 15. products of chitosan or chitin, which (had)……which has

Line 20. Please write H2O2 as H2O2 here and elsewhere

Line 23. Please write the full form of DPPH, O2-, H2O2, FRAP, BHA at their first use.

Introduction

Line 42. Please write Forsythia suspense in italics

Line 44. …which can use as natural----which can be used as natural

Material and Methods

Line 70. Better describe the methods in brief on how the analysis and calculation is done

Line 79-80. The supplementation level of COS following former research in our lab (data was not shown).----- The supplementation level of COS was based on our previous preliminary research (Do you mean to say this?)

Results , Discussion

English grammar/ sentences are not correctly written and not easily understandable. Therefore a thorough check for the English is recommended by professional English speaker.

Thanks

Author Response

Comments and Suggestions for Authors

The manuscript describes about dietary supplementation of Chitosan oligosaccharide for having its ability to alleviate oxidative stress in rats that are exposed to hydrogen peroxide which is quite interesting.

The English of this paper needs corrections. Please have the whole paper checked again and corrected by an English-speaking person who is familiar with scientific writing.

Author response: Thanks for your suggestion. We agree with you, and we ask help from native speakers. Please See Line 12-15, 20-23, 27-28, 32-37, 94, 121-128, 170-171, 182-187, 209-212, 216-218, 227-234, 240-247, 252-254, 263-267 and 270-274. Thanks a lot.

Some of the sentences are started with abbreviations. Do not start the sentences with abbreviations.

Author response: Thanks for your suggestion. We agree with you, and make changes. Please See Line 21-23 and 52. Thanks a lot.

Specific comments

Abstract

Line 13. Supplementation antioxidants – Supplementation of antioxidants

Author response: Thanks for your suggestion. We make changes. Please See Line 12-13. Thanks a lot.

Line 15. products of chitosan or chitin, which (had)……which has

Author response: Thanks for your suggestion. We make changes. Please See Line 14-15. Thanks a lot.

Line 20. Please write H2O2 as H2O2 here and elsewhere

Author response: Thanks for your suggestion. all H2O2 has changed to H2O2, please see line 17, 19, 22, 24, 26, and 46.

Line 23. Please write the full form of DPPH, O2-, H2O2, FRAP, BHA at their first use.

 Author response: Thanks for your suggestion. We make changes. Please See Line 22-23. Thanks a lot.

Introduction

 Line 42. Please write Forsythia suspense in italics

Author response: Thanks for your suggestion. We make changes. Please See Line 39. Thanks a lot.

 Line 44. …which can use as natural----which can be used as natural

 Author response: Thanks for your suggestion. We make changes. Please See Line 42. Thanks a lot.

Material and Methods

 Line 70. Better describe the methods in brief on how the analysis and calculation is done

Author response: Thanks for your suggestion. We make changes. Please See Line 65-85. Thanks a lot.

 Line 79-80. The supplementation level of COS following former research in our lab (data was not shown).----- The supplementation level of COS was based on our previous preliminary research (Do you mean to say this?)

Author response: Thanks for your suggestion. We make changes. Please See Line 94. Thanks a lot.

Results , Discussion

 English grammar/ sentences are not correctly written and not easily understandable. Therefore a thorough check for the English is recommended by professional English speaker.

Author response: Thanks for your suggestion. We agree with you, and made changes. Please See Line 121-128, 170-171, 182-187, 209-212, 216-218, 227-234, 240-247, 252-254, 263-267 and 270-274. Thanks a lot.

Reviewer 2 Report

Comment for author: This study evaluates the effects of the dietary supplementation with COS alleviates the damage caused by oxidative stress in Sprague Dawley rats challenged with hydrogen peroxide. The topic is interesting and topical.

Major concerns:

To clarify the aim and usefulness of the study more information in the introduction and material & methods sections are needed. It seems essential that the authors describe more precise what is novel for the present study compare to the already conducted studies. Results section, Line 121-135, needs to be rewritten since it is difficult to read. Summarize the results in a more concise and understandable way. change “H2O2” into H2O2 To improve the English language of the manuscript the authors should check with a native English speaker. 

Author Response

Comments and Suggestions for Authors

To clarify the aim and usefulness of the study more information in the introduction and material & methods sections are needed. It seems essential that the authors describe more precise what is novel for the present study compare to the already conducted studies. Results section, Line 121-135, needs to be rewritten since it is difficult to read. Summarize the results in a more concise and understandable way. change “H2O2” into H2O2 To improve the English language of the manuscript the authors should check with a native English speaker. 

Author response: Thanks for your suggestion. We agree with you, and we ask help from native speakers. Please See Line 12-15, 20-23, 27-28, 32-37, 94, 121-128, 170-171, 182-187, 209-212, 216-218, 227-234, 240-247, 252-254, 263-267 and 270-274. Meanwhile, all H2O2 has changed to H2O2, please see line 17, 19, 22, 24, 26, and 46. Thanks a lot.

Reviewer 3 Report

This manuscript is well-written and the result is interesting. I only have one point about the figure one: why no err bars?

Author Response

Comments and Suggestions for Authors

This manuscript is well-written and the result is interesting. I only have one point about the figure one: why no err bars?

Author response: Thanks for your suggestion. We agree with you, and we made change. Please See Line 129, Figure 1. Thanks a lot.